# ONCE-FOR-ALL: TRAIN ONE NETWORK AND SPECIALIZE IT FOR EFFICIENT DEPLOYMENT

**Han Cai[1], Chuang Gan[2], Tianzhe Wang[1], Zhekai Zhang[1], Song Han[1]**
[1]Massachusetts Institute of Technology, [2]MIT-IBM Watson AI Lab
{hancai, chuangg, songhan}@mit.edu

## ABSTRACT

We address the challenging problem of efficient inference across many devices and resource constraints, especially on edge devices. Conventional approaches either manually design or use neural architecture search (NAS) to find a specialized neural network and train it from scratch for *each* case, which is computationally prohibitive (causing $CO_2$ emission as much as 5 cars' lifetime Strubell et al. (2019)) thus unscalable. In this work, we propose to train a once-for-all (OFA) network that supports diverse architectural settings by decoupling training and search, to reduce the cost. We can quickly get a specialized sub-network by selecting from the OFA network without additional training. To efficiently train OFA networks, we also propose a novel progressive shrinking algorithm, a generalized pruning method that reduces the model size across many more dimensions than pruning (depth, width, kernel size, and resolution). It can obtain a surprisingly large number of sub-networks ($> 10^{19}$) that can fit different hardware platforms and latency constraints while maintaining the same level of accuracy as training independently. On diverse edge devices, OFA consistently outperforms state-of-the-art (SOTA) NAS methods (up to 4.0% ImageNet top1 accuracy improvement over MobileNetV3, or same accuracy but $1.5\times$ faster than MobileNetV3, $2.6\times$ faster than EfficientNet w.r.t measured latency) while reducing many orders of magnitude GPU hours and $CO_2$ emission. In particular, OFA achieves a new SOTA 80.0% ImageNet top-1 accuracy under the mobile setting (<600M MACs). OFA is the winning solution for the 3rd Low Power Computer Vision Challenge (LPCVC), DSP classification track and the 4th LPCVC, both classification track and detection track. Code and 50 pre-trained models (for many devices & many latency constraints) are released at https://github.com/mit-han-lab/once-for-all.

## 1 INTRODUCTION

Deep Neural Networks (DNNs) deliver state-of-the-art accuracy in many machine learning applications. However, the explosive growth in model size and computation cost gives rise to new challenges on how to efficiently deploy these deep learning models on *diverse* hardware platforms, since they have to meet *different* hardware efficiency constraints (e.g., latency, energy). For instance, one mobile application on App Store has to support a diverse range of hardware devices, from a high-end Samsung Note10 with a dedicated neural network accelerator to a 5-year-old Samsung S6 with a much slower processor. With different hardware resources (e.g., on-chip memory size, #arithmetic units), the optimal neural network architecture varies significantly. Even running on the same hardware, under different battery conditions or workloads, the best model architecture also differs a lot.

Given different hardware platforms and efficiency constraints (defined as deployment scenarios), researchers either design compact models specialized for mobile (Howard et al., 2017; Sandler et al., 2018; Zhang et al., 2018) or accelerate the existing models by compression (Han et al., 2016; He et al., 2018) for efficient deployment. However, designing specialized DNNs for every scenario is engineer-expensive and computationally expensive, either with human-based methods or NAS. Since such methods need to *repeat* the network design process and *retrain* the designed network from scratch for *each* case. Their total cost grows linearly as the number of deployment scenarios increases, which will result in excessive energy consumption and $CO_2$ emission (Strubell et al., 2019). It makes them unable to handle the vast amount of hardware devices (23.14 billion IoT devices till

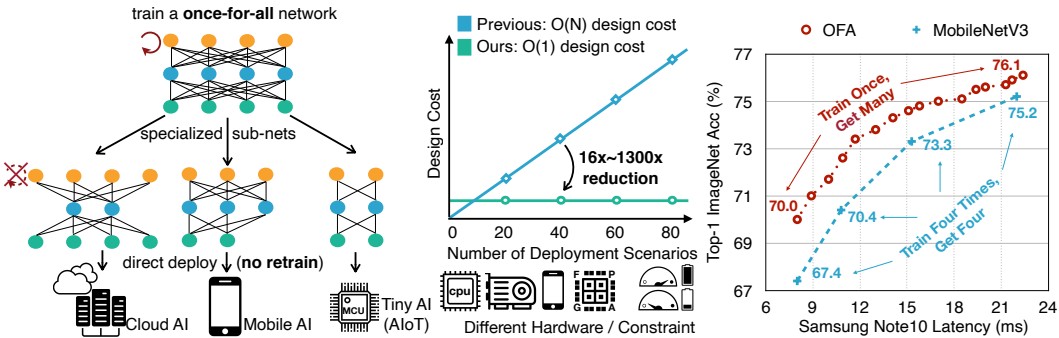

Figure 1: Left: a single once-for-all network is trained to support versatile architectural configurations including depth, width, kernel size, and resolution. Given a deployment scenario, a specialized sub-network is directly selected from the once-for-all network without training. Middle: this approach reduces the cost of specialized deep learning deployment from O(N) to O(1). Right: once-for-all network followed by model selection can derive many accuracy-latency trade-offs by training only once, compared to conventional methods that require repeated training.

2018[1]) and highly dynamic deployment environments (different battery conditions, different latency requirements, etc.).

This paper introduces a new solution to tackle this challenge – designing a *once-for-all network* that can be directly deployed under diverse architectural configurations, amortizing the training cost. The inference is performed by selecting only part of the once-for-all network. It flexibly supports different depths, widths, kernel sizes, and resolutions without retraining. A simple example of *Once-for-All* (OFA) is illustrated in Figure 1 (left). Specifically, we decouple the model training stage and the neural architecture search stage. In the model training stage, we focus on improving the accuracy of all sub-networks that are derived by selecting different parts of the once-for-all network. In the model specialization stage, we sample a subset of sub-networks to train an accuracy predictor and latency predictors. Given the target hardware and constraint, a predictor-guided architecture search (Liu et al., 2018) is conducted to get a specialized sub-network, and the cost is negligible. As such, we reduce the total cost of specialized neural network design from O(N) to O(1) (Figure 1 middle).

However, training the once-for-all network is a non-trivial task, since it requires joint optimization of the weights to maintain the accuracy of a large number of sub-networks (more than $10^{19}$ in our experiments). It is computationally prohibitive to enumerate all sub-networks to get the exact gradient in each update step, while randomly sampling a few sub-networks in each step will lead to significant accuracy drops. The challenge is that different sub-networks are interfering with each other, making the training process of the whole once-for-all network inefficient. To address this challenge, we propose a *progressive shrinking* algorithm for training the once-for-all network. Instead of directly optimizing the once-for-all network from scratch, we propose to first train the largest neural network with *maximum* depth, width, and kernel size, then progressively fine-tune the once-for-all network to support *smaller* sub-networks that share weights with the larger ones. As such, it provides better initialization by selecting the most important weights of larger sub-networks, and the opportunity to distill smaller sub-networks, which greatly improves the training efficiency. From this perspective, progressive shrinking can be viewed as a generalized network pruning method that shrinks multiple dimensions (depth, width, kernel size, and resolution) of the full network rather than only the width dimension. Besides, it targets on maintaining the accuracy of all sub-networks rather than a single pruned network.

We extensively evaluated the effectiveness of OFA on ImageNet with many hardware platforms (CPU, GPU, mCPU, mGPU, FPGA accelerator) and efficiency constraints. Under all deployment scenarios, OFA consistently improves the ImageNet accuracy by a significant margin compared to SOTA hardware-aware NAS methods while saving the GPU hours, dollars, and $CO_2$ emission by orders of magnitude. On the ImageNet mobile setting (less than 600M MACs), OFA achieves a new SOTA 80.0% top1 accuracy with 595M MACs (Figure 2). To the best of our knowledge, this is the first time that the SOTA ImageNet top1 accuracy reaches 80% under the mobile setting.

---

[1]https://www.statista.com/statistics/471264/iot-number-of-connected-devices-worldwide/

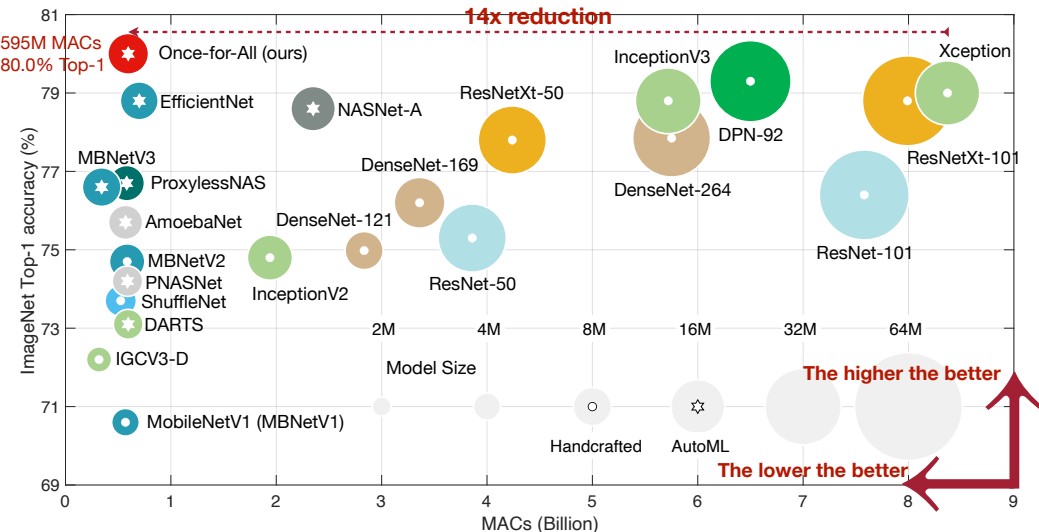

Figure 2: Comparison between OFA and state-of-the-art CNN models on ImageNet. OFA provides 80.0% ImageNet top1 accuracy under the mobile setting (< 600M MACs).

## 2 RELATED WORK

**Efficient Deep Learning.** Many efficient neural network architectures are proposed to improve the hardware efficiency, such as SqueezeNet (Iandola et al., 2016), MobileNets (Howard et al., 2017; Sandler et al., 2018), ShuffleNets (Ma et al., 2018; Zhang et al., 2018), etc. Orthogonal to architecting efficient neural networks, model compression (Han et al., 2016) is another very effective technique for efficient deep learning, including network pruning that removes redundant units (Han et al., 2015) or redundant channels (He et al., 2018; Liu et al., 2017), and quantization that reduces the bit width for the weights and activations (Han et al., 2016; Courbariaux et al., 2015; Zhu et al., 2017).

**Neural Architecture Search.** Neural architecture search (NAS) focuses on automating the architecture design process (Zoph & Le, 2017; Zoph et al., 2018; Real et al., 2019; Cai et al., 2018a; Liu et al., 2019). Early NAS methods (Zoph et al., 2018; Real et al., 2019; Cai et al., 2018b) search for high-accuracy architectures without taking hardware efficiency into consideration. Therefore, the produced architectures (e.g., NASNet, AmoebaNet) are not efficient for inference. Recent hardware-aware NAS methods (Cai et al., 2019; Tan et al., 2019; Wu et al., 2019) directly incorporate the hardware feedback into architecture search. Hardware-DNN co-design techniques (Jiang et al., 2019b;a; Hao et al., 2019) jointly optimize neural network architectures and hardware architectures. As a result, they can improve inference efficiency. However, given new inference hardware platforms, these methods need to repeat the architecture search process and retrain the model, leading to prohibitive GPU hours, dollars, and $CO_2$ emission. They are not scalable to a large number of deployment scenarios. The individually trained models do not share any weight, leading to large total model size and high downloading bandwidth.

**Dynamic Neural Networks.** To improve the efficiency of a given neural network, some work explored skipping part of the model based on the input image. For example, Wu et al. (2018); Liu & Deng (2018); Wang et al. (2018) learn a controller or gating modules to adaptively drop layers; Huang et al. (2018) introduce early-exit branches in the computation graph; Lin et al. (2017) adaptively prune channels based on the input feature map; Kuen et al. (2018) introduce stochastic downsampling point to reduce the feature map size adaptively. Recently, Slimmable Nets (Yu et al., 2019; Yu & Huang, 2019b) propose to train a model to support multiple width multipliers (e.g., 4 different global width multipliers), building upon existing human-designed neural networks (e.g., MobileNetV2 0.35, 0.5, 0.75, 1.0). Such methods can adaptively fit different efficiency constraints at runtime, however, still inherit a pre-designed neural network (e.g., MobileNet-v2), which limits the degree of flexibility (e.g., only width multiplier can adapt) and the ability in handling new deployment scenarios where the pre-designed neural network is not optimal. In this work, in contrast, we enable a much more diverse architecture space (depth, width, kernel size, and resolution) and a significantly larger number of architectural settings ($10^{19}$ v.s. 4 (Yu et al., 2019)). Thanks to the diversity and the large design

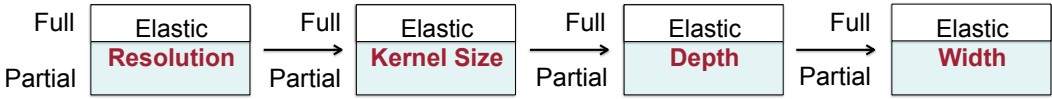

Figure 3: Illustration of the progressive shrinking process to support different depth $D$, width $W$, kernel size $K$ and resolution $R$. It leads to a large space comprising diverse sub-networks ($> 10^{19}$).

space, we can derive new specialized neural networks for many different deployment scenarios rather than working on top of an existing neural network that limits the optimization headroom. However, it is more challenging to train the network to achieve this flexibility, which motivates us to design the progressive shrinking algorithm to tackle this challenge.

## 3 METHOD

### 3.1 PROBLEM FORMALIZATION

Assuming the weights of the once-for-all network as $W_o$ and the architectural configurations as $\{arch_i\}$, we then can formalize the problem as

$$\min_{W_o} \sum_{arch_i} \mathcal{L}_{val}\big(C(W_o, arch_i)\big), \tag{1}$$

where $C(W_o, arch_i)$ denotes a selection scheme that selects part of the model from the once-for-all network $W_o$ to form a sub-network with architectural configuration $arch_i$. The overall training objective is to optimize $W_o$ to make each supported sub-network maintain the *same* level of accuracy as *independently* training a network with the same architectural configuration.

### 3.2 ARCHITECTURE SPACE

Our once-for-all network provides one model but supports many sub-networks of different sizes, covering four important dimensions of the convolutional neural networks (CNNs) architectures, i.e., depth, width, kernel size, and resolution. Following the common practice of many CNN models (He et al., 2016; Sandler et al., 2018; Huang et al., 2017), we divide a CNN model into a sequence of units with gradually reduced feature map size and increased channel numbers. Each unit consists of a sequence of layers where only the first layer has stride 2 if the feature map size decreases (Sandler et al., 2018). All the other layers in the units have stride 1.

We allow each unit to use arbitrary numbers of layers (denoted as *elastic depth*); For each layer, we allow to use arbitrary numbers of channels (denoted as *elastic width*) and arbitrary kernel sizes (denoted as *elastic kernel size*). In addition, we also allow the CNN model to take arbitrary input image sizes (denoted as *elastic resolution*). For example, in our experiments, the input image size ranges from 128 to 224 with a stride 4; the depth of each unit is chosen from $\{2, 3, 4\}$; the width expansion ratio in each layer is chosen from $\{3, 4, 6\}$; the kernel size is chosen from $\{3, 5, 7\}$. Therefore, with 5 units, we have roughly $((3 \times 3)^2 + (3 \times 3)^3 + (3 \times 3)^4)^5 \approx 2 \times 10^{19}$ different neural network architectures and each of them can be used under 25 different input resolutions. Since all of these sub-networks share the same weights (i.e., $W_o$) (Cheung et al., 2019), we only require 7.7M parameters to store all of them. Without sharing, the total model size will be prohibitive.

### 3.3 TRAINING THE ONCE-FOR-ALL NETWORK

**Naïve Approach.** Training the once-for-all network can be cast as a multi-objective problem, where each objective corresponds to one sub-network. From this perspective, a naïve training approach is to directly optimize the once-for-all network from scratch using the exact gradient of the overall objective, which is derived by enumerating all sub-networks in each update step, as shown in Eq. (1). The cost of this approach is linear to the number of sub-networks. Therefore, it is only applicable to scenarios where a limited number of sub-networks are supported (Yu et al., 2019), while in our case, it is computationally prohibitive to adopt this approach.

Another naïve training approach is to sample a few sub-networks in each update step rather than enumerate all of them, which does not have the issue of prohibitive cost. However, with such a large number of sub-networks that share weights, thus interfere with each other, we find it suffers from

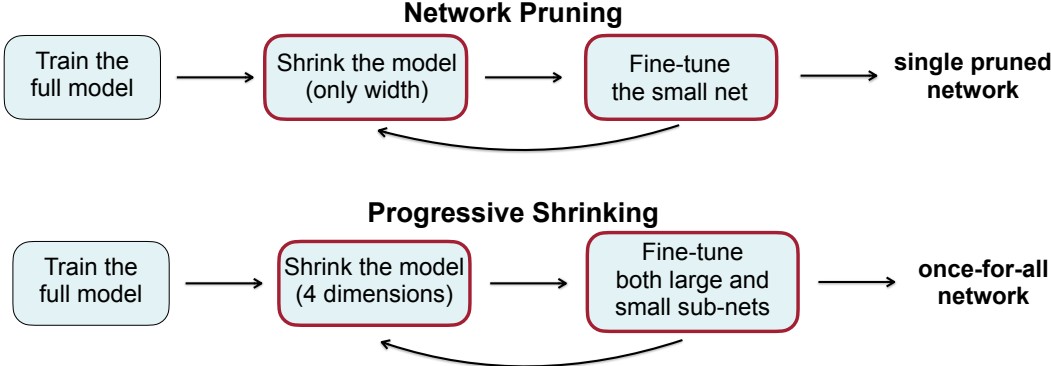

Figure 4: Progressive shrinking can be viewed as a generalized network pruning technique with much higher flexibility. Compared to network pruning, it shrinks more dimensions (not only width) and provides a much more powerful once-for-all network that can fit different deployment scenarios rather than a single pruned network.

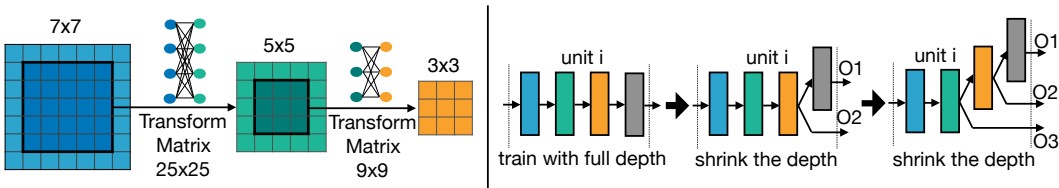

Figure 5: Left: Kernel transformation matrix for elastic kernel size. Right: Progressive shrinking for elastic depth. Instead of skipping each layer independently, we keep the first $D$ layers and skip the last $(4 - D)$ layers. The weights of the early layers are shared.

significant accuracy drop. In the following section, we introduce a solution to address this challenge, i.e., *progressive shrinking*.

**Progressive Shrinking.** The once-for-all network comprises many sub-networks of different sizes where small sub-networks are nested in large sub-networks. To prevent interference between the sub-networks, we propose to enforce a training order from large sub-networks to small sub-networks in a progressive manner. We name this training scheme as *progressive shrinking* (PS). An example of the training process with PS is provided in Figure 3 and Figure 4, where we start with training the largest neural network with the maximum kernel size (e.g., 7), depth (e.g., 4), and width (e.g., 6). Next, we progressively fine-tune the network to support smaller sub-networks by gradually adding them into the sampling space (larger sub-networks may also be sampled). Specifically, after training the largest network, we first support elastic kernel size which can choose from {3, 5, 7} at each layer, while the depth and width remain the maximum values. Then, we support elastic depth and elastic width sequentially, as is shown in Figure 3. The resolution is elastic throughout the whole training process, which is implemented by sampling different image sizes for each batch of training data. We also use the knowledge distillation technique after training the largest neural network (Hinton et al., 2015; Ashok et al., 2018; Yu & Huang, 2019b). It combines two loss terms using both the soft labels given by the largest neural network and the real labels.

Compared to the naïve approach, PS prevents small sub-networks from interfering large sub-networks, since large sub-networks are already well-trained when the once-for-all network is fine-tuned to support small sub-networks. Regarding the small sub-networks, they share the weights with the large ones. Therefore, PS allows initializing small sub-networks with the most important weights of well-trained large sub-networks, which expedites the training process. Compared to network pruning (Figure 4), PS also starts with training the full model, but it shrinks not only the width dimension but also the depth, kernel size, and resolution dimensions of the full model. Additionally, PS fine-tunes both large and small sub-networks rather than a single pruned network. As a result, PS provides a much more powerful once-for-all network that can fit diverse hardware platforms and efficiency constraints compared to network pruning. We describe the details of the PS training flow as follows:

Figure 6: Progressive shrinking for elastic width. In this example, we progressively support 4, 3, and 2 channel settings. We perform channel sorting and pick the most important channels (with large L1 norm) to initialize the smaller channel settings. The important channels' weights are shared.

- **Elastic Kernel Size** (Figure 5 left). We let the center of a 7x7 convolution kernel also serve as a 5x5 kernel, the center of which can also be a 3x3 kernel. Therefore, the kernel size becomes elastic. The challenge is that the centering sub-kernels (e.g., 3x3 and 5x5) are shared and need to play multiple roles (independent kernel and part of a large kernel). The weights of centered sub-kernels may need to have different distribution or magnitude as different roles. Forcing them to be the same degrades the performance of some sub-networks. Therefore, we introduce kernel transformation matrices when sharing the kernel weights. We use separate kernel transformation matrices for different layers. Within each layer, the kernel transformation matrices are shared among different channels. As such, we only need $25 \times 25 + 9 \times 9 = 706$ extra parameters to store the kernel transformation matrices in each layer, which is negligible.

- **Elastic Depth** (Figure 5 right). To derive a sub-network that has $D$ layers in a unit that originally has $N$ layers, we keep the *first* D layers and skip the last $N - D$ layers, rather than keeping *any* $D$ layers as done in current NAS methods (Cai et al., 2019; Wu et al., 2019). As such, one depth setting only corresponds to one combination of layers. In the end, the weights of the first D layers are shared between large and small models.

- **Elastic Width** (Figure 6). Width means the number of channels. We give each layer the flexibility to choose different channel expansion ratios. Following the progressive shrinking scheme, we first train a full-width model. Then we introduce a channel sorting operation to support partial widths. It reorganizes the channels according to their importance, which is calculated based on the L1 norm of a channel's weight. Larger L1 norm means more important. For example, when shrinking from a 4-channel-layer to a 3-channel-layer, we select the largest 3 channels, whose weights are shared with the 4-channel-layer (Figure 6 left and middle). Thereby, smaller sub-networks are initialized with the most important channels on the once-for-all network which is already well trained. This channel sorting operation preserves the accuracy of larger sub-networks.

## 3.4 Specialized Model Deployment with Once-for-all Network

Having trained a once-for-all network, the next stage is to derive the specialized sub-network for a given deployment scenario. The goal is to search for a neural network that satisfies the efficiency (e.g., latency, energy) constraints on the target hardware while optimizing the accuracy. Since OFA decouples model training from neural architecture search, we do not need any training cost in this stage. Furthermore, we build *neural-network-twins* to predict the latency and accuracy given a neural network architecture, providing a quick feedback for model quality. It eliminates the repeated search cost by substituting the measured accuracy/latency with predicted accuracy/latency (twins).

Specifically, we randomly sample 16K sub-networks with different architectures and input image sizes, then measure their accuracy on 10K validation images sampled from the original training set. These [architecture, accuracy] pairs are used to train an accuracy predictor to predict the accuracy of a model given its architecture and input image size[2]. We also build a latency lookup table (Cai et al., 2019) on each target hardware platform to predict the latency. Given the target hardware and latency constraint, we conduct an evolutionary search (Real et al., 2019) based on the neural-network-twins to get a specialized sub-network. Since the cost of searching with neural-network-twins is negligible,

---

[2]Details of the accuracy predictor is provided in Appendix A.

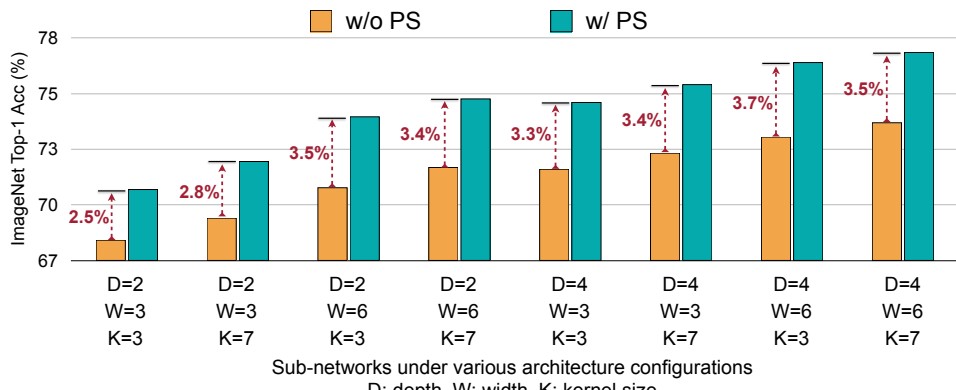

Figure 7: ImageNet top1 accuracy (%) performances of sub-networks under resolution $224 \times 224$. "(D = $d$, W = $w$, K = $k$)" denotes a sub-network with $d$ layers in each unit, and each layer has an width expansion ratio $w$ and kernel size $k$.

we only need 40 GPU hours to collect the data pairs, and the cost stays constant regardless of #deployment scenarios.

## 4 EXPERIMENTS

In this section, we first apply the progressive shrinking algorithm to train the once-for-all network on ImageNet (Deng et al., 2009). Then we demonstrate the effectiveness of our trained once-for-all network on various hardware platforms (Samsung S7 Edge, Note8, Note10, Google Pixel1, Pixel2, LG G8, NVIDIA 1080Ti, V100 GPUs, Jetson TX2, Intel Xeon CPU, Xilinx ZU9EG, and ZU3EG FPGAs) with different latency constraints.

### 4.1 TRAINING THE ONCE-FOR-ALL NETWORK ON IMAGENET

**Training Details.** We use the same architecture space as MobileNetV3 (Howard et al., 2019). For training the full network, we use the standard SGD optimizer with Nesterov momentum 0.9 and weight decay $3e^{-5}$. The initial learning rate is 2.6, and we use the cosine schedule (Loshchilov & Hutter, 2016) for learning rate decay. The full network is trained for 180 epochs with batch size 2048 on 32 GPUs. Then we follow the schedule described in Figure 3 to further fine-tune the full network[3]. The whole training process takes around 1,200 GPU hours on V100 GPUs. This is a one-time training cost that can be amortized by many deployment scenarios.

**Results.** Figure 7 reports the top1 accuracy of sub-networks derived from the once-for-all networks that are trained with our progressive shrinking (PS) algorithm and without PS respectively. Due to space limits, we take 8 sub-networks for comparison, and each of them is denoted as "(D = $d$, W = $w$, K = $k$)". It represents a sub-network that has $d$ layers for all units, while the expansion ratio and kernel size are set to $w$ and $k$ for all layers. PS can improve the ImageNet accuracy of sub-networks by a significant margin under all architectural settings. Specifically, without architecture optimization, PS can achieve 74.8% top1 accuracy using 226M MACs under the architecture setting (D=4, W=3, K=3), which is on par with MobileNetV3-Large. In contrast, without PS, it only achieves 71.5%, which is 3.3% lower.

### 4.2 SPECIALIZED SUB-NETWORKS FOR DIFFERENT HARDWARE AND CONSTRAINTS

We apply our trained once-for-all network to get different specialized sub-networks for diverse hardware platforms: from the cloud to the edge. **On cloud devices**, the latency for GPU is measured with batch size 64 on NVIDIA 1080Ti and V100 with Pytorch 1.0+cuDNN. The CPU latency is measured with batch size 1 on Intel Xeon E5-2690 v4+MKL-DNN. **On edge devices**, including mobile phones, we use Samsung, Google and LG phones with TF-Lite, batch size 1; for mobile GPU,

---

[3]Implementation details can be found in Appendix B.

| Model | ImageNet Top1 (%) | MACs | Mobile latency | Search cost (GPU hours) | Training cost (GPU hours) | Total cost ($N = 40$) | | |
|---|---|---|---|---|---|---|---|---|
| | | | | | | GPU hours | $CO_2$e (lbs) | AWS cost |
| MobileNetV2 [31] | 72.0 | 300M | 66ms | 0 | $150N$ | 6k | 1.7k | $18.4k |
| MobileNetV2 #1200 | 73.5 | 300M | 66ms | 0 | $1200N$ | 48k | 13.6k | $146.9k |
| NASNet-A [44] | 74.0 | 564M | - | $48,000N$ | - | 1,920k | 544.5k | $5875.2k |
| DARTS [25] | 73.1 | 595M | - | $96N$ | $250N$ | 14k | 4.0k | $42.8k |
| MnasNet [33] | 74.0 | 317M | 70ms | $40,000N$ | - | 1,600k | 453.8k | $4896.0k |
| FBNet-C [36] | 74.9 | 375M | - | $216N$ | $360N$ | 23k | 6.5k | $70.4k |
| ProxylessNAS [4] | 74.6 | 320M | 71ms | $200N$ | $300N$ | 20k | 5.7k | $61.2k |
| SinglePathNAS [8] | 74.7 | 328M | - | $288 + 24N$ | $384N$ | 17k | 4.8k | $52.0k |
| AutoSlim [38] | 74.2 | 305M | 63ms | 180 | $300N$ | 12k | 3.4k | $36.7k |
| MobileNetV3-Large [15] | 75.2 | 219M | 58ms | - | $180N$ | 7.2k | 1.8k | $22.2k |
| OFA w/o PS | 72.4 | 235M | 59ms | 40 | 1200 | 1.2k | 0.34k | $3.7k |
| OFA w/ PS | **76.0** | 230M | 58ms | 40 | 1200 | 1.2k | 0.34k | $3.7k |
| OFA w/ PS #25 | 76.4 | 230M | 58ms | 40 | $1200 + 25N$ | 2.2k | 0.62k | $6.7k |
| OFA w/ PS #75 | **76.9** | 230M | 58ms | 40 | $1200 + 75N$ | 4.2k | 1.2k | $13.0k |
| OFA$_{Large}$ w/ PS #75 | **80.0** | 595M | - | 40 | $1200 + 75N$ | 4.2k | 1.2k | $13.0k |

Table 1: Comparison with SOTA hardware-aware NAS methods on Pixel1 phone. OFA decouples model training from neural architecture search. The search cost and training cost both stay constant as the number of deployment scenarios grows. "#25" denotes the specialized sub-networks are fine-tuned for 25 epochs after grabbing weights from the once-for-all network. "$CO_2e$" denotes $CO_2$ emission which is calculated based on Strubell et al. (2019). AWS cost is calculated based on the price of on-demand P3.16xlarge instances.

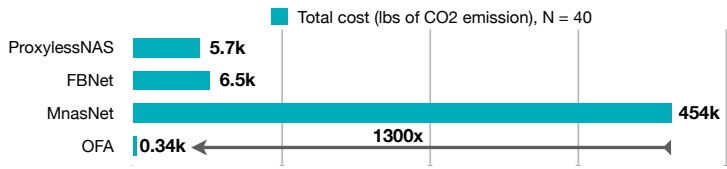

Figure 8: OFA saves orders of magnitude design cost compared to NAS methods.

we use Jetson TX2 with Pytorch 1.0+cuDNN, batch size of 16; for embedded FPGA, we use Xilinx ZU9EG and ZU3EG FPGAs with Vitis AI[4], batch size 1.

**Comparison with NAS on Mobile Devices.** Table 1 reports the comparison between OFA and state-of-the-art hardware-aware NAS methods on the mobile phone (Pixel1). OFA is much more efficient than NAS when handling multiple deployment scenarios since the cost of OFA is *constant* while others are *linear* to the number of deployment scenarios ($N$). **With $N = 40$, the total $CO_2$ emissions of OFA is 16× fewer than ProxylessNAS, 19× fewer than FBNet, and 1,300× fewer than MnasNet (Figure 8).** Without retraining, OFA achieves 76.0% top1 accuracy on ImageNet, which is 0.8% higher than MobileNetV3-Large while maintaining similar mobile latency. We can further improve the top1 accuracy to 76.4% by fine-tuning the specialized sub-network for 25 epochs and to 76.9% by fine-tuning for 75 epochs. Besides, we also observe that OFA with PS can achieve 3.6% better accuracy than without PS.

**OFA under Different Computational Resource Constraints.** Figure 9 summarizes the results of OFA under different MACs and Pixel1 latency constraints. OFA achieves 79.1% ImageNet top1 accuracy with 389M MACs, being 2.8% more accurate than EfficientNet-B0 that has similar MACs. With 595M MACs, OFA reaches a new SOTA 80.0% ImageNet top1 accuracy under the mobile setting (<600M MACs), which is 0.2% higher than EfficientNet-B2 while using 1.68× fewer MACs. More importantly, OFA runs much faster than EfficientNets on hardware. Specifically, with 143ms Pixel1 latency, OFA achieves 80.1% ImageNet top1 accuracy, being 0.3% more accurate and 2.6× faster than EfficientNet-B2. We also find that training the searched neural architectures from scratch cannot reach the same level of accuracy as OFA, suggesting that not only neural architectures but also pre-trained weights contribute to the superior performances of OFA.

Figure 10 reports detailed comparisons between OFA and MobileNetV3 on six mobile devices. Remarkably, **OFA can produce the entire trade-off curves with many points over a wide range of latency constraints by training only once** (green curve). It is impossible for previous NAS methods (Tan et al., 2019; Cai et al., 2019) due to the prohibitive training cost.

---

[4]https://www.xilinx.com/products/design-tools/vitis/vitis-ai.html

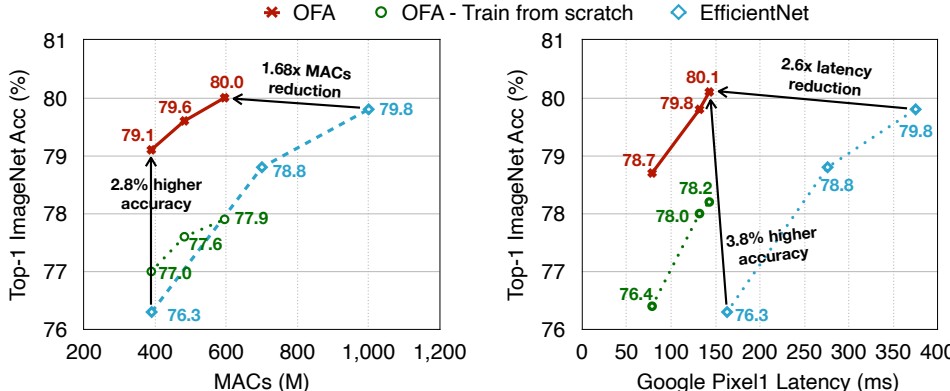

Figure 9: OFA achieves 80.0% top1 accuracy with 595M MACs and 80.1% top1 accuracy with 143ms Pixel1 latency, setting a new SOTA ImageNet top1 accuracy on the mobile setting.

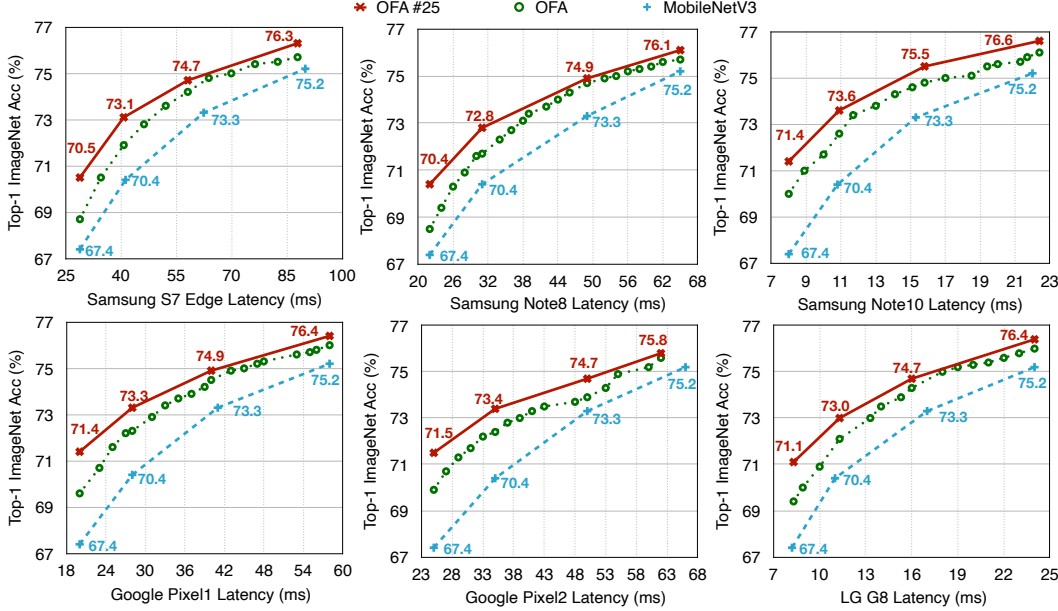

Figure 10: OFA consistently outperforms MobileNetV3 on mobile platforms.

**OFA for Diverse Hardware Platforms.** Besides the mobile platforms, we extensively studied the effectiveness of OFA on six additional hardware platforms (Figure 11) using the ProxylessNAS architecture space (Cai et al., 2019). OFA consistently improves the trade-off between accuracy and latency by a significant margin, especially on GPUs which have more parallelism. With similar latency as MobileNetV2 0.35, "OFA #25" improves the ImageNet top1 accuracy from MobileNetV2's 60.3% to 72.6% (+12.3% improvement) on the 1080Ti GPU. Detailed architectures of our specialized models are shown in Figure 14. It reveals the insight that using the *same* model for different deployment scenarios with *only* the width multiplier modified has a limited impact on efficiency improvement: the accuracy drops quickly as the latency constraint gets tighter.

**OFA for Specialized Hardware Accelerators.** There has been plenty of work on NAS for general-purpose hardware, but little work has been focused on specialized hardware accelerators. We quantitatively analyzed the performance of OFA on two FPGAs accelerators (ZU3EG and ZU9EG) using Xilinx Vitis AI with 8-bit quantization, and discuss two design principles.

**Principle 1**: memory access is expensive, computation is cheap. An efficient CNN should do *as much as* computation with *a small amount* of memory footprint. The ratio is defined as the arithmetic intensity (OPs/Byte). The higher OPs/Byte, the less memory bounded, the easier to parallelize. Thanks to OFA's diverse choices of sub-network architectures ($10^{19}$) (Section 3.3), and the OFA

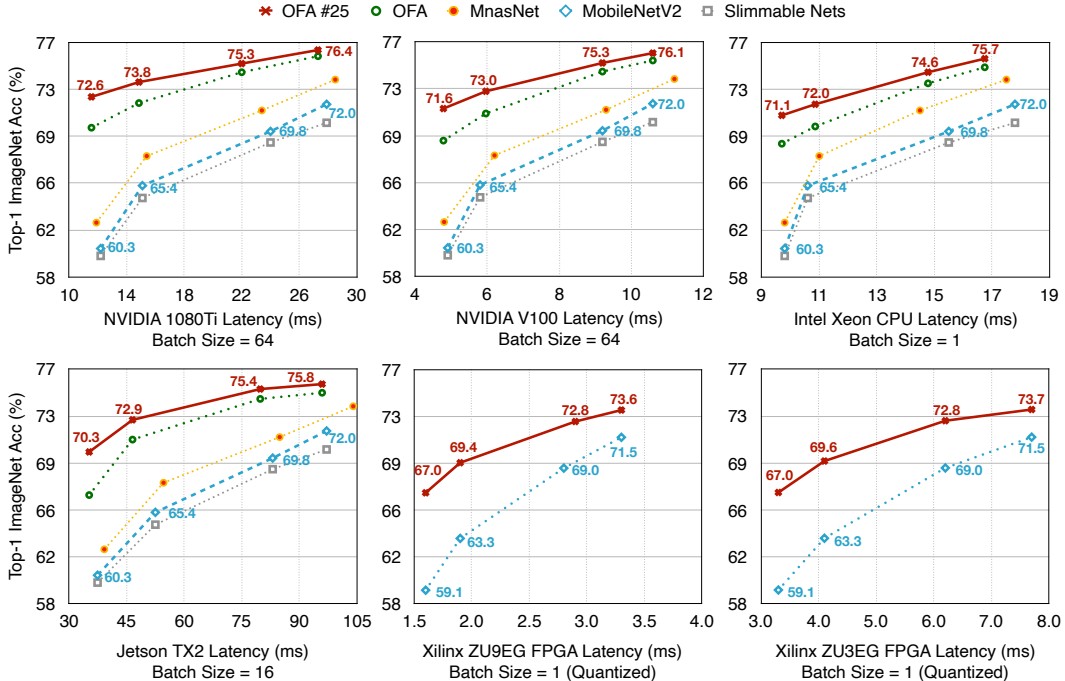

Figure 11: Specialized OFA models consistently achieve significantly higher ImageNet accuracy with similar latency than non-specialized neural networks on CPU, GPU, mGPU, and FPGA. More remarkably, specializing for a new hardware platform does not add training cost using OFA.

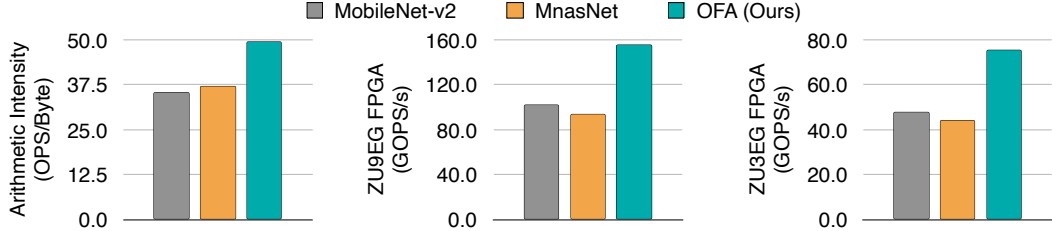

Figure 12: OFA models improve the arithmetic intensity (OPS/Byte) and utilization (GOPS/s) compared with the MobileNetV2 and MnasNet (measured results on Xilinx ZU9EG and ZU3EG FPGA).

model twin that can quickly give the accuracy/latency feedback (Section 3.4), the evolutionary search can automatically find a CNN architecture that has higher arithmetic intensity. As shown in Figure 12, OFA's arithmetic intensity is 48%/43% higher than MobileNetV2 and MnasNet (MobileNetV3 is not supported by Xilinx Vitis AI). Removing the memory bottleneck results in higher utilization and GOPS/s by 70%-90%, pushing the operation point to the upper-right in the roofline model (Williams et al., 2009), as shown in Figure 13. (70%-90% looks small in the log scale but that is significant).

**Principle 2**: the CNN architecture should be co-designed with the hardware accelerator's cost model. The FPGA accelerator has a specialized depth-wise engine that is pipelined with the point-wise engine. The pipeline throughput is perfectly matched for 3x3 kernels. As a result, OFA's searched model only has 3x3 kernel (Figure 14, a) on FPGA, despite 5x5 and 7x7 kernels are also in the search space. Additionally, large kernels sometimes cause "out of BRAM" error on FPGA, giving high cost. On Intel Xeon CPU, however, more than 50% operations are large kernels. Both FPGA and GPU models are wider than CPU, due to the large parallelism of the computation array.

## 5 CONCLUSION

We proposed *Once-for-All* (OFA), a new methodology that decouples model training from architecture search for efficient deep learning deployment under a large number of hardware platforms. Unlike

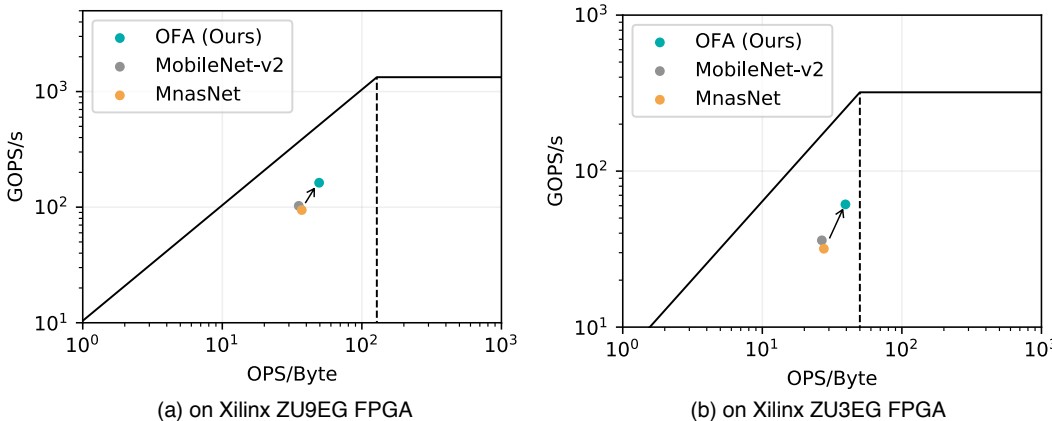

Figure 13: Quantative study of OFA's roofline model on Xilinx ZU9EG and ZU3EG FPGAs (log scale). OFA model increased the arithmetic intensity by 33%/43% and GOPS/s by 72%/92% on these two FPGAs compared with MnasNet.

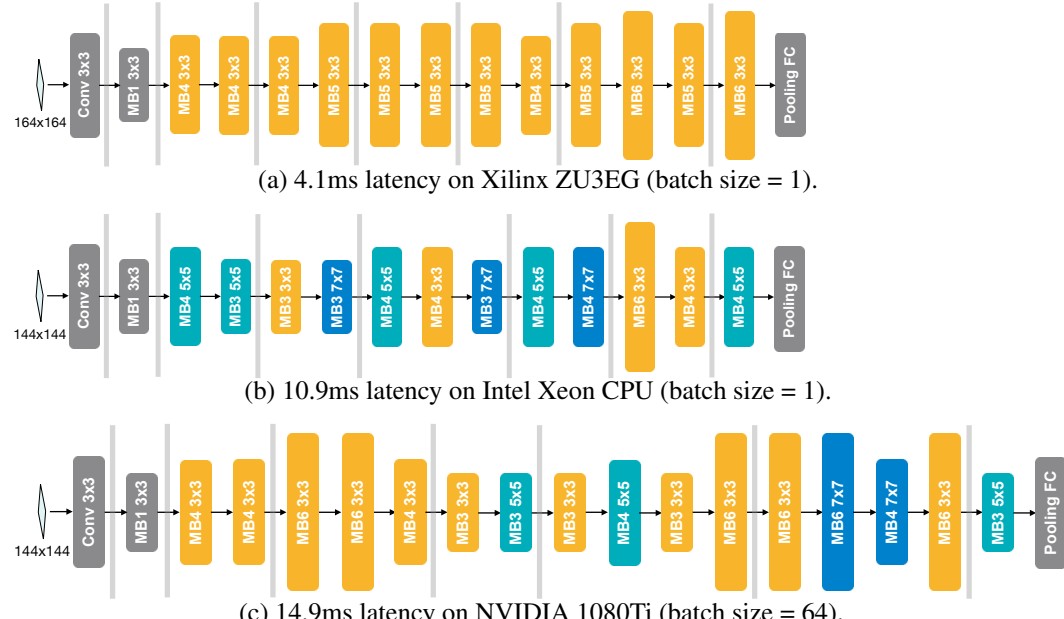

Figure 14: OFA can design specialized models for different hardware and different latency constraint. "MB4 3x3" means "mobile block with expansion ratio 4, kernel size 3x3". FPGA and GPU models are wider than CPU model due to larger parallelism. Different hardware has different cost model, leading to different optimal CNN architectures. OFA provides a unified and efficient design methodology.

previous approaches that design and train a neural network for *each* deployment scenario, we designed a *once-for-all network* that supports different architectural configurations, including elastic depth, width, kernel size, and resolution. It reduces the training cost (GPU hours, energy consumption, and $CO_2$ emission) by orders of magnitude compared to conventional methods. To prevent sub-networks of different sizes from interference, we proposed a progressive shrinking algorithm that enables a large number of sub-network to achieve the same level of accuracy compared to training them independently. Experiments on a diverse range of hardware platforms and efficiency constraints demonstrated the effectiveness of our approach. OFA provides an automated ecosystem to efficiently design efficient neural networks with the hardware cost model in the loop.

ACKNOWLEDGMENTS

We thank NSF Career Award #1943349, MIT-IBM Watson AI Lab, Google-Daydream Research Award, Samsung, Intel, Xilinx, SONY, AWS Machine Learning Research Award for supporting this

research. We thank Samsung, Google and LG for donating mobile phones. We thank Shuang Wu and Lei Deng for drawing the Figure 2.

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

## A   DETAILS OF THE ACCURACY PREDICTOR

We use a three-layer feedforward neural network that has 400 hidden units in each layer as the accuracy predictor. Given a model, we encode each layer in the neural network into a one-hot vector based on its kernel size and expand ratio, and we assign zero vectors to layers that are skipped. Besides, we have an additional one-hot vector that represents the input image size. We concatenate these vectors into a large vector that represents the whole neural network architecture and input image size, which is then fed to the three-layer feedforward neural network to get the predicted accuracy. In our experiments, this simple accuracy prediction model can provide very accurate predictions. At convergence, the root-mean-square error (RMSE) between predicted accuracy and estimated accuracy on the test set is only 0.21%. Figure 15 shows the relationship between the RMSE of the accuracy prediction model and the final results (i.e., the accuracy of selected sub-networks). We can find that lower RMSE typically leads to better final results.

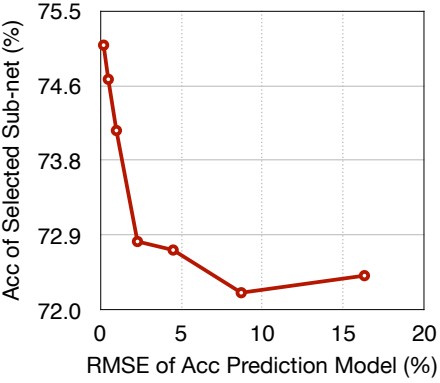

Figure 15: Performances of selected sub-networks using different accuracy prediction model.

## B   IMPLEMENTATION DETAILS OF PROGRESSIVE SHRINKING

After training the full network, we first have one stage of fine-tuning to incorporate elastic kernel size. In this stage (i.e., $K \in [7, 5, 3]$), we sample one sub-network in each update step. The network is fine-tuned for 125 epochs with an initial learning rate of 0.96. All other training settings are the same as training the full network.

Next, we have two stages of fine-tuning to incorporate elastic depth. We sample two sub-networks and aggregate their gradients in each update step. The first stage (i.e., $D \in [4, 3]$) takes 25 epochs with an initial learning rate of 0.08 while the second stage (i.e., $D \in [4, 3, 2]$) takes 125 epochs with an initial learning rate of 0.24.

Finally, we have two stages of fine-tuning to incorporate elastic width. We sample four sub-networks and aggregate their gradients in each update step. The first stage (i.e., $W \in [6, 4]$) takes 25 epochs with an initial learning rate of 0.08 while the second stage (i.e., $W \in [6, 4, 3]$) takes 125 epochs with an initial learning rate of 0.24.

