# OpenReview forum: "Once-for-All: Train One Network and Specialize it for Efficient Deployment"
_ICLR.cc/2020/Conference — Accept (Poster)_

### Official Review · AnonReviewer2 · 2019-10-15
**Official Blind Review #2**

**Rating:** 6

**Review:**

In this papers, the authors learn a Once-for-all net. This starts as a big neural network which is trained normally (albeit with input images of different resolutions). It is then fine-tuned while sampling sub-networks with progressively smaller kernels, then lower depth, then width (while still sampling larger networks occasionally, as it reads). This results in a network from which one can extract sub-networks for various resource constraints (latency, memory etc.) that perform well without a need for retraining.

This paper is well written, and the results are very good. However there are serious problems that need addressing.

The method as described *is not reproducible*. The scheduling of sampling subnetworks is alluded to on page 4, and that's it. It is essential that the authors include their exact subnet sampling schedule e.g. as pseudocode with hyperparameters. There is no point doing good work if other researchers cannot build off it.

On another reproducibility note, as far as I can tell, the original model isn't given. There would be no harm in adding this to the appendix.

Figure 1 is misleading, as we don't find out until later in the paper that Once For All #25 means that each of these points was finetuned for a further 25 epochs (which on ImageNet is non-trivial). This defeats the narrative of the paper (once-for-all plus some fine-tuning isn't exactly once-for-all).

Is there a reason why the progressive shrinking goes resolution->kernel->depth->width? Was this just the permutation that worked best? I would be curious as to why this is.

For elastic width, I wasn't sure why the "channel sorting operation preserves the accuracy of larger sub-networks". Could you please elaborate?

Kudos on adding CO2 emissions in Table 2, I hope this gets reported more often.

In the introduction, the authors talk about iPhones and then the hardware considered is Samsung and Google. A minor note, but it seems inconsistent.

Another minor note, in Table 2, (Strubell et al) should be out of the brackets, as it is part of the sentence.

Given that there are 10^19 subnetworks that can be sampled, it would be nice to see more than 3-4 appear on a plot. This makes it seem like they might have been cherry-picked. Sampling a few 100/1000 subnets and producing some Pareto curves would be both interesting and insightful.

Pros
-------
- Good results
- Well written
- Neat idea

Cons
-------
- Training details are obfuscated. This paper should not be accepted without them.
- Very few subnetworks of the vast quantity that exist are observed.

In conclusion, I am giving this paper a weak reject, as it is currently impossible to reproduce, and as such, is of no use to the community. If the authors remedy this I will gladly raise my score.

**Experience Assessment:**

I have published one or two papers in this area.

**Review Assessment: Checking Correctness Of Derivations And Theory:**

N/A

**Review Assessment: Checking Correctness Of Experiments:**

I assessed the sensibility of the experiments.

**Review Assessment: Thoroughness In Paper Reading:**

I read the paper thoroughly.

---

> ### Author Response · Authors · 2019-11-10
> **Our response to Reviewer #2**
>
> Thanks very much for your constructive and detailed comments. We will fix the typos and remove “Once for All #25” from figure 1.
> 1. Training details and code release.
> For reproduction, we will add a detailed and clear description of our training details by Nov. 15. We are also working on cleaning the code. The training code and pre-trained models will be posted anonymously in the OpenReview by Nov. 22.
>
> 2. Sample more sub-networks and produce some Pareto curves.
> That’s a great idea. Thanks for the suggestion. We will update our figures showing the entire trade-off curves rather than a few points by Nov. 15.
>
> 3. Adding the original model in the appendix.
> Thank you for the suggestion. We will add a figure showing the detailed architecture of the full (original) model in the appendix.
>
> 4. Why the progressive shrinking goes resolution->kernel->depth->width.
> The order is determined based on the difficulty of each task. Intuitively, we hope the model to complete easy tasks first and then handle more difficult tasks, similar to the idea of curriculum learning.
>
> 5. Why the channel sorting operation preserves the accuracy of larger sub-networks.
> When performing the channel sorting operation on a specific layer, we first sort the input dimension of the layer according to their importance (i.e., L1 norm). Then the output dimension of the previous layer is reorganized accordingly to make sure the functionality of large sub-networks does not change.
>
> We have also summarized all of our planned updates in our general response above. If there are any additional comments on the paper or on the planned updates, please don’t hesitate to let us know.

---

### Official Review · AnonReviewer3 · 2019-10-23
**Official Blind Review #3**

**Rating:** 6

**Review:**

This paper tries to tackle the problem of searching best architectures for specialized resource constraint deployment scenarios. The authors basically take a two-step approach: First train a large network including all the small networks with weight sharing and some specially designed trick (e.g., progressive shrinking). Second, use prediction based NAS method to learn the performance/inference prediction module, from which the good sub architecture corresponding to a particular scenario is obtained. The experiments show that the proposed method is promising.

Pros:


1. It is an interesting new paradigm that tries to solve AutoML for different deployment scenarios “once for all”.  AFAIK there is no prior works thinking in this way.
2. It is useful and encouraging to see the proposed method achieves satisfactory performances, on par with the current best method specially designed for different deployment environment, while the computational cost is reduced by a large margin.
3. Paper is clearly written and easy to understand.

Cons:

1. The motivation towards “progressive shrinking (PS)” is not that clear. It seems natural to train a large network, and from it to train sub structures, since overparameterization helps NN training. However, it is hard to imagine that training from large to small could eliminate the “interfering” of subnetworks, let alone “while maintaining the same accuracy as independently trained networks”.  To me it is neither theoretically nor empirically supported (Please note the training of subnetworks definitely affect the learnt weights of the big one through weight sharing). In particular, the subnetworks with weight sharing could achieve the same, or even better performances compared with those non shared counterparts, which seems too good to be true.
    1. A possible explanation might be that the overparameterization brings additional gain in the optimization process of each small network, especially with the help of knowledge distillation. If that is true, an additional ablation study should be done to separate the benefits of PS, and the disadvantage of weight sharing (i.e., interfering).
2. I see no statements about code release. If a clear, and TIMELY code (for the SEARCH phase, not only for the Eval phase) release could be done, then at least from the perspective of application, the impact of this paper could be further enhanced.



**Experience Assessment:**

I have published one or two papers in this area.

**Review Assessment: Checking Correctness Of Derivations And Theory:**

I assessed the sensibility of the derivations and theory.

**Review Assessment: Checking Correctness Of Experiments:**

I assessed the sensibility of the experiments.

**Review Assessment: Thoroughness In Paper Reading:**

I read the paper at least twice and used my best judgement in assessing the paper.

---

> ### Author Response · Authors · 2019-11-10
> **Our response to Reviewer #3**
>
> Thanks very much for your constructive comments.
> 1. Why training from large to small can prevent interference between sub-networks.
> Training large sub-networks can also benefit small sub-networks to learn useful features. For example, after finishing the step of elastic kernel size, the sub-network (D=3, W=6, K=7, R=224) can already achieve 69.1% top-1 accuracy on ImageNet without any fine-tuning. This is consistent with previous observations in network pruning [1,2,3]. By training from large to small, both large sub-networks and small sub-networks can reuse previously learned knowledge (or features). Empirically, we find that it is helpful for the optimization of the shared weights with the goal of supporting large sub-networks and small sub-networks at the same time.
>
> 2. Why subnetworks with weight sharing could achieve the same, or even better performances compared with those non shared counterparts.
> We first want to clarify that we are not targeting at improving the accuracy of a specific sub-network for a **single** scenario; instead, we want to improve the accuracy-efficiency trade-off on **many** hardware platforms while reducing the total training cost. To avoid confusion about the goal of this paper, we will emphasize our main contribution and make it more clear in the revision.
>
> We conjecture the reason for this result is that smaller sub-networks can benefit from getting the knowledge transferred from well-trained large sub-networks through inheriting weights from large sub-networks and knowledge distillation.
>
> Regarding separating the benefits of PS and the disadvantage of weight sharing (i.e., interfering), we want to clarify that weight sharing is an essential component of the OFA framework since it is prohibitive to download and store so many networks independently on resource-constrained edge devices.
>
> 3. Code release.
> Thank you for the suggestion. We definitely hope this work can be a useful tool for application purposes. We are currently cleaning the code. The training code and pre-trained models will be released anonymously in the OpenReview by Nov. 22.
>
> We have also summarized all of our planned updates in our general response above. If there are any additional comments on the paper or on the planned updates, please don’t hesitate to let us know.
>
> [1] Han, Song, et al. "Deep compression: Compressing deep neural networks with pruning, trained quantization and huffman coding." in ICLR 2016.
> [2] Liu, Zhuang, et al. "Learning efficient convolutional networks through network slimming." in ICCV 2017.
> [3] He, Yihui, et al. "Channel pruning for accelerating very deep neural networks." in ICCV 2017.

---

### Official Review · AnonReviewer1 · 2019-10-23
**Official Blind Review #1**

**Rating:** 6

**Review:**

In this manuscript, authors propose an OFA NAS framework. They train a supernet first and then finetune the elastic version of the large network. After training, the sub-networks derived from the supernet can be applied for different scenarios directly without retraining. The motivation is clear and interesting. My concerns are as follows.
1.	When sampling sub-networks, a prediction model is applied to predict the accuracy of networks. It is interesting to show the accuracy of the prediction model itself and how it will influence the final selection.
2.	The results compared in Table 2 are outdated. Authors should at least add the result of MobileNetV3.

**Experience Assessment:**

I have read many papers in this area.

**Review Assessment: Checking Correctness Of Derivations And Theory:**

I assessed the sensibility of the derivations and theory.

**Review Assessment: Checking Correctness Of Experiments:**

I assessed the sensibility of the experiments.

**Review Assessment: Thoroughness In Paper Reading:**

I read the paper at least twice and used my best judgement in assessing the paper.

---

> ### Author Response · Authors · 2019-11-10
> **Our response to Reviewer #1**
>
> Thanks very much for your constructive comments.
> 1. Performance of the accuracy prediction model and how it influences the final selection.
> We will add a figure in the appendix by Nov. 15, showing the relationship between the performance of the accuracy prediction model and the accuracy of selected sub-networks.
>
> 2. Comparison to MobileNetV3 in Table 2.
> Thanks for the suggestion. We agree that it is essential to compare our model to MobileNetV3 which gives the current SOTA performances on mobile platforms. To have an Apple-to-Apple comparison with it, we will apply our method to the same architecture space as MobileNetV3. The new results will be included by Nov. 15.
>
> We have also summarized all of our planned updates in our general response above. If there are any additional comments on the paper or on the planned updates, please don’t hesitate to let us know.

---

### Public Comment · ~Rudy_Chin1 · 2019-10-09
**Interesting work**

Dear authors,

This is a really interesting work that one can train a large network such that the sub-networks within the large network still work really well, which appears to be even better than the individual trained ones!

I'm trying to implement this idea and would like to know the specific hyper-parameters used in the paper. Specifically, how long do you train the large network before switching to the elastic version. In training the elastic version, do you still train the large network? If so, how are their losses combined? When training the elastic version gradually, what is the specific learning rate since the paper mentioned that you train with small learning rate so that the large network wouldn't deviate too much from the pre-trained weights. Also, can you elaborate on how the number of epochs and learning rate used to fine-tune each of the elastic space chosen and how they affect the results?

Another question I have is that you mentioned using distillation for the large network to distill the sub-networks, do you do the same for the independent trained models? Specifically, in Table 1, do you use knowledge distillation for the independent trained models? If not, I think it is not clear if the proposed OFA network indeed produce sub-networks that outperform the individually trained ones.

Thanks,
Rudy

---

> ### Author Response · Authors · 2019-10-18
> **Open Source & Motivation**
>
> Hi Rudy,
>
> Thanks for your interest. We will release the code after the double-blind review.
>
> We use 150 epochs to train the full (large) network before switching to the elastic version. In training the elastic version, the full (large) network is not trained. We set the learning rate for fine-tuning as 0.04 (1/10 initial learning rate). We choose the hyper-parameters by cross-validation (learning rates around 0.04 give stable results; increasing the number of epochs can usually improve the results).
>
> Regarding your second question, we do not claim that OFA produces sub-networks that outperform the individually trained ones.  Our main contribution is to reduce the total cost of handling **many** deployment scenarios (hardware platforms and constraints), which is crucial for real-world applications, rather than targeting a **single** scenario. Therefore, the key advantage of OFA is that OFA can efficiently specialize for different deployment scenarios while individually trained models cannot. Even independently training the sub-network with distillation (using the same teacher network as OFA), the accuracy slightly improved from 74.3% to 74.7%, which is still at the same level as OFA produced sub-network (74.8%).
>
> Best,
> Authors

---

### Public Comment · ~Jason_Kuen1 · 2019-11-07
**Interesting work and a missing reference**

Hi, thanks for the interesting work that advances the progress of efficient deep learning.

I especially find the idea of progressive shrinking intriguing. It is said that the smaller sub-networks distill knowledge from larger sub-networks.. Since all sub-networks share the same weights, wouldn't training smaller sub-networks change the prediction behavior of larger sub-networks (making them unreliable for distillation)?

There is a missing reference to a related work that also similarly focuses on multiple efficiency configurations using a single model without retraining. It would be good if the authors could acknowledge it.
- Stochastic Downsampling for Cost-Adjustable Inference and Improved Regularization in Convolutional Networks, CVPR 2018.

---

> ### Author Response · Authors · 2019-11-10
> **Thanks for suggesting a related paper. We will add a reference to the paper in the revision.**
>
> Hi Jason,
>
> Regarding your question about distillation, the teacher model does not share weights with the OFA network. Specifically, after training the full network, one copy of the full network weights is used as the teacher model, and another copy of the full network weights is used for further training to support smaller sub-networks. Therefore, training smaller sub-networks does not affect the teacher model.
>
> Best,
> Authors

---

### Author Response · Authors · 2019-11-10
**Our general response**

We sincerely thank all reviewers for their comments. We summarize our planned updates as follows:

1. We will apply our method to the same architecture space as MobileNetV3. The new results will be included by Nov. 15.

2. We will add a figure in the appendix by Nov. 15, showing the relationship between the performance of the accuracy prediction model and the accuracy of selected sub-networks.

3. We will update our figures showing the entire trade-off curves with many points (rather than a few points) of OFA on different hardware platforms by Nov. 15.

4. We will add the detailed architecture of the full model in the appendix.

5. For reproduction, we will include a detailed description of our training details by Nov. 15. We are working on cleaning the code. The training code and pre-trained models will be released anonymously in the OpenReview by Nov. 22.

If there are any additional comments on the paper or on the planned updates, please don’t hesitate to let us know.

---

### Author Response · Authors · 2019-11-15
**Revision Uploaded**

We sincerely thank all reviewers for their constructive comments. We have revised our paper accordingly with the promised results and implementation details included. Please check out the new version!

Our pre-trained model and training code are available at:
https://drive.google.com/open?id=1GrLufnGc_3UYG6l7kBX3JYjUqPr8ZaUQ

1. We updated the experiment section (Section 4) with the new results in the MobileNetV3 search space. OFA consistently outperforms MobileNetV3 on various mobile platforms and latency constraints.

2. In Appendix A, we included a figure showing the relationship between the performance of the accuracy prediction model and the accuracy of selected sub-networks.

3. We updated figure1 and included a new figure (figure 5) that shows the entire trade-off curves of OFA on mobile platforms.

4. In Appendix C, we added a table showing the detailed architecture of the full network.

5. In Appendix E, we included implementation details of the progressive shrinking algorithm.

If there are any additional comments on the paper or on the code, please don’t hesitate to let us know.

---

### Decision · Program_Chairs · 2019-12-19

**Decision:**

Accept (Poster)

**Comment:**

The authors propose a new method for neural architecture search, except it's not exactly that because model training is separated from architecture, which is the main point of the paper. Once this network is trained, sub-networks can be distilled from it and used for specific tasks.

The paper as submitted missed certain details, but after this was pointed out by reviewers the details were satisfactorily described by the authors.

The idea of the paper is original and interesting. The paper is correct and, after the revisions by authors, complete. In my view, this is sufficient for acceptance.